# Phi 6 Bacteriophage Inactivation by Metal Salts, Metal Powders, and Metal Surfaces

**DOI:** 10.3390/v14020204

**Published:** 2022-01-21

**Authors:** Katja Molan, Ramin Rahmani, Daniel Krklec, Miha Brojan, David Stopar

**Affiliations:** 1Department of Microbiology, Biotechnical Faculty, University of Ljubljana, Večna pot 111, 1000 Ljubljana, Slovenia; Katja.molan@bf.uni-lj.si (K.M.); Daniel.Krklec@gmail.com (D.K.); 2Department of Mechanical and Industrial Engineering, Tallinn University of Technology, Ehitajate tee 5, 19086 Tallinn, Estonia; Ramin.Rahmaniahranjani@taltech.ee; 3Laboratory for Nonlinear Mechanics, Faculty of Mechanical Engineering, University of Ljubljana, Aškerčeva cesta 6, 1000 Ljubljana, Slovenia; Miha.Brojan@fs.uni-lj.si

**Keywords:** bacteriophage (phage) Phi6, metal salts, ceramic–metal powders, metals, composite materials

## Abstract

The interaction of phages with abiotic environmental surfaces is usually an understudied field of phage ecology. In this study, we investigated the virucidal potential of different metal salts, metal and ceramic powders doped with Ag and Cu ions, and newly fabricated ceramic and metal surfaces against Phi6 bacteriophage. The new materials were fabricated by spark plasma sintering (SPS) and/or selective laser melting (SLM) techniques and had different surface free energies and infiltration features. We show that inactivation of Phi6 in solutions with Ag and Cu ions can be as effective as inactivation by pH, temperature, or UV. Adding powder to Ag and Cu ion solutions decreased their virucidal effect. The newly fabricated ceramic and metal surfaces showed very good virucidal activity. In particular, 45%TiO_2_ + 5%Ag + 45%ZrO_2_ + 5%Cu, in addition to virus adhesion, showed virucidal and infiltration properties. The results indicate that more than 99.99% of viruses deposited on the new ceramic surface were inactivated or irreversibly attached to it.

## 1. Introduction

Viruses can spread and contaminate surfaces by aerosols and droplets. The lack of effective virucidal strategies can often be due to a poor understanding of virus stability on different surfaces. In this work, we used the model Phi6 bacteriophage and checked its infectivity in contact with metal ions, metal and ceramic powders, as well as on different engineered surfaces. Phi6 virus has been extensively used as a surrogate for the study of enveloped RNA animal viruses [1,2,3,4,5], including SARS coronaviruses [6,7]. Similar to SARS-CoV-2, it is enveloped by a lipid membrane, has spike proteins, and is of similar size (80–100 nm) [8,9,10,11,12].

Phi 6 stability to environmental factors in suspension, such as temperature, pH, RH, and light, has been determined [2,7,11,12,13,14,15,16,17,18]. Much less, however, is known about phage stability in the presence of metal ions, metal powders, and ceramic or metal surfaces. For example, the virucidal effect of copper ions has been described against Phi6 as well as against influenza virus, norovirus, monkeypox, vaccinia virus, human immunodeficiency virus (HIV), SARS-CoV, and SARS-CoV-2 [19,20,21,22,23,24,25,26,27,28]. Silver nanoparticles, on the other hand, have been shown to inhibit herpes simplex virus, human parainfluenza virus type 3, vaccinia virus, respiratory syncytial virus, Tacaribe virus, and hepatitis B virus [29,30,31,32,33]. Surprisingly, the mechanisms of action are still not well understood [34]. It has been shown that by binding to the surface of HIV-1 glycoproteins, silver nanoparticles destroy disulfide bonds [35]. Such destruction of viral surface glycoproteins has also been suggested for virucidal activity against influenza virus [36,37]. There is one study of the silver nanoparticles effect on Phi6 bacteriophage [38], whereas no studies have been performed on the effect of other metal ions (i.e., Fe, Al, Cr, Co Mn, Ni) on Phi6 stability.

Although silver and copper ions have well-demonstrated virucidal properties, their use as pure metal surfaces is limited due to their high price. In contrast, ceramic surfaces are much more affordable but suffer from brittleness [39,40]. By doping ceramics with metal powders, ceramic brittleness can be reduced significantly [41,42]. The virucidal properties of silver nanoparticles incorporated in ceramic or metal lattices are still largely unexplored. Therefore, new composite ceramic and metal materials with tailored mechanical properties that retain self-cleaning properties are in high demand. For example, silver can be incorporated into metal alloys as nanoparticles with well-defined shape, particle size, and polydispersity, which dictate their biocidal activities and toxicity [43]. Recently, Ag nanoparticles were successfully integrated into TiO_2_/Ag ceramics and Ti6Al4V–TiO_2_/Ag composites using combined SPS and/or SLM techniques [44]. This manufacturing route is promising for the production of solidified samples for in vitro and in vivo antibacterial and antiviral applications.

The adhesion of viruses to surfaces can significantly change their stability. The stability of attached viruses can either increase or decrease. For example, the stability of enteric viruses in contact with different soft surfaces was affected by surface type and environment conditions [45,46]. A recent study found that the stability of SARS-CoV-2 was also surface-dependent. The longest stability was observed on plastic and stainless-steel surfaces (up to 72 h after application), while on copper, no infectible SARS-CoV-2 was detected after 4 h [27]. On the other hand, much shorter persistence was observed on copper surfaces, with no infective viruses determined after 4 h [47]. Similarly, silver solid surfaces can inhibit viruses by binding to and interacting with viral surface proteins or by denaturing enzymes when reacting with amino, carboxyl, imidazole, and sulfhydryl groups [48,49]. The effect of metal alloys and composite ceramic surfaces on virus survival is much less researched.

In this work, we investigated the virucidal potential of different metal salts (AgNO_3_, FeSO_4_, Al_2_(SO_4_)_3_, NiCl_2_, K_2_Cr_2_O_7_, CuSO_4_), powders of titanium and zirconium oxides doped with Ag and Cu (TiO_2_–ZrO_2_–Ag–Cu, TiO_2_–Ag–Cu), and ceramic and metal surfaces (TiO_2_ + 10%Ag + 10%Cu, 45%TiO_2_ + 5%Ag + 45%ZrO_2_ + 5%Cu, steel, Co28Cr6Mo). We fabricated new disk materials by spark plasma sintering (SPS) and/or selective laser melting (SLM) techniques and determined the surface free energy. The virucidal activity of the new materials against bacteriophage Phi6 was compared to the effect of physicochemical parameters (i.e., T, pH, RH, UV, ultrasound) on phage infectivity. The results showed that phage titers can significantly decrease already after 15 min of treatment with different metal salt solutions, ceramic powders, or fabricated surfaces. The new fabricated ceramic and metal composite surfaces showed some unexpected virucidal properties such as infiltration with more than 99.99% removal of infective virus particles.

## 2. Materials and Methods

All virucidal experiments were performed with an overnight culture of *Pseudomonas* sp. (DSM 21482) and bacteriophage Phi6 obtained from Leibnitz-Institut (DSMZ-Deutsche Sammlung von Mikroorganismen und Zellkulturen GmbH, Braunschweig, Germany). Bacteria were cultured in Tryptic Soy Broth (TSB) medium at 28 °C and 180 rpm, overnight. For phage enrichment, 1% of the overnight bacterial culture was added to fresh TSB medium and grown to the exponential phase. Next, 1 mL of exponential-phase bacteria and 300 µL of Phi6 phage suspension were added to 30 mL of SM buffer, followed by incubation overnight at 50 rpm and 28 °C as described previously [50]. The enriched phage suspensions were centrifuged (13,000 rpm, 4 °C, 10 min), and the supernatant filtered through a 0.22 µm filter. To obtain a stock viral suspension with a titer of 10^10^–10^11^, the phage suspension was centrifuged at 100,000× *g* for 2 h at 4 °C. The phage stock was stored at 4 °C in SM buffer (7.5 pH) until further use. The virucidal effect was determined in terms of Plaque-Forming Units (PFU) before and after exposure of the virus particles for 15 min to different conditions/solutions/suspensions/material surfaces using the double-layer agar overlay plaque method [51]. Viral titer was calculated as [(PFU/volume of phage)*phage dilution]. All experiments were performed independently at least three times.

### 2.1. Testing Virus Stability to Physicochemical Factors

To determine viral titer after exposure of Phi6 to different physicochemical factors, phage suspensions were diluted in Phosphate-Buffered Saline (PBS, 7.4 pH), and viral titer was determined by the double-layer agar method [51]. To test the effect of temperature, 10 µL of phage suspension was added to 490 µL of PBS buffer in a microcentrifuge tube, mixed, and incubated for 15 min at different temperatures in a PCR cycler (Biometra Tone Series, Analytik Jena GmbH, Jena, Germany). To test the effect of pH, solutions with a pH ranging from 2 to 12.5 were prepared. To 490 µL of a solution with known pH, 10 µL of phage suspension was added, and the suspension was incubated for 15 min at room temperature without mixing under light conditions. The effects of relative humidity (RH) values were studied in a constant climatic chamber (Memmert, Schwabach, Germany) at 4 different temperatures (10, 19, 28, and 33 °C) and 2 RH values (10% and 80%). In this experiment, 10 µL of the phage suspension was added to a microtiter plate and incubated for 24 h at given RH and temperature. Next, 100 µL of PBS solution was added to the wells and mixed with a pipette, and the suspension was transferred to a microcentrifuge tube containing 400 µL of PBS buffer to test for viral titer. To test the effects of ultrasound on phage stability, microcentrifuge tubes containing 490 µL of PBS and 10 µL of phage suspension were placed on ice for 10 min and then sonicated with a sonotrode pulse (3 mm probe at 12 microns of amplitude, 150 Watt Ultrasonic Disintegrator Mk2, MSE Scientific Instruments, Heathfield, UK) from 30 to 100 s. The effect of UV radiation was tested in an Ultraviolet Crosslinker chamber (Analytik Jena GmbH, Jena, Germany) in a 12-well titer plate containing 490 µL of PBS and 10 µL of phage suspension. Each column (3 wells) of the titer plate was irradiated with a different amount of energy, from 100 mJ/cm^2^ to 1000 mJ/cm^2^.

### 2.2. Testing the Virucidal Effect of Metal Ion Salts

Stock solutions of silver nitrate, iron sulphate, aluminum sulphate, copper sulphate, and nickel chloride (II) were prepared as 0.5 M solutions in Milli-Q water (5.5 pH). Metal salt concentrations in the range from 0.1 M to 0.001 M were prepared in MQ water. To 10 µL of the stock phage suspension, 490 µL of the solution with the tested metal ion concentration was added, and the mixture was incubated for 15 min at room temperature (without mixing, under light conditions). The virus titer was determined based on PFU using the double-layer agar method.

### 2.3. Testing the Virucidal Effect of Metal and Ceramic Powders

TiO_2_–ZrO_2_–Ag–Cu, TiO_2_–Ag–Cu, and Co28Cr6Mo powders were prepared as 0.5 M stock suspensions of the respective dopant metal ions in MQ water (5.5 pH). We prepared powder suspensions ranging from 0.001 to 0.25 M by diluting with Milli-Q water. To ensure better solubility, the powder suspensions were sonicated and vortexed before use. To 10 µL of the stock phage suspension, 490 µL of the tested metal and ceramic powder suspension was added, and the mixture was incubated for 15 min at room temperature (without mixing, under light conditions). To determine the virus titer, the solutions containing the phages were serially diluted, and the number of PFU was determined using the double-layer agar method.

### 2.4. Testing the Virucidal Effect on Composite Disks Surfaces

The disks were prepared by spark plasma sintering (SPS, FCT Systeme GmbH, Effelder-Rauenstein, Germany) in vacuum chamber with a nitrogen flow. SPS is the most popular, advanced, and rapid sintering process in powder metallurgy (PM), which for metal alloys, metal matrix composites, and ceramics guarantees the highest consolidation and densification percentage. During SPS, a low-voltage pulsed electric current is imposed on the electrodes. Electrical discharges pass through the material powders which are enclosed inside a graphite mold. The combination of samples, pressure, sintering time, and temperature has a pivotal role in PM procedures. In our experiments approximately 15 g of each mixture was plasma-sintered for 20 min at a 50 MPa pressure. Temperature was 800 °C for TiO_2_ + 10%Ag + 10%Cu, 45%TiO_2_ + 5%Ag + 45%ZrO_2_ + 5%Cu ceramic disks, and 1000 °C for pre-alloyed gas-atomized Co28Cr6Mo metallic discs.

The protocol to test the virucidal effect on disk surfaces is schematically depicted in Figure 1. Five droplets containing 10 µL of phage stock solution were placed on the disk surface and incubated for 15 min at room temperature (non-mixing, light conditions), after which the liquid was removed, and virus titer determined. After one min, the surface was swabbed with a cotton swab soaked in 100 µL PBS buffer and additionally with a dry cotton swab to collect the viruses on the surface completely. Both the moist and the dry swab were placed in a microcentrifuge tube containing 400 µL PBS and vigorously vortexed for 10 s to detach the virus particles from the swabs. Next, 100 µL of undiluted or 500 µL of diluted PBS virus suspensions were used to infect bacterial cells and to determine viral titers. To determine the residual viruses left on the swabs after detachment, both cotton swabs were placed on TSA agar plates and doused with soft agar containing an inoculum of bacteria. The TSA plates with cotton swabs were photographed and visually compared. Although swabbing removed most of the virus from the disk surface, strongly attached viruses could remain on the surface. To determine strongly attached infective viruses, the disks were sonicated in 1–5 mL of PBS buffer for 5 min in a sonication bath (ASonic Ultrasonic Cleaner, degas mode, water temperature 22 °C). In the next step, 500 µL of PBS solution with detached virus particles was used for viral titer determination. We checked independently the effect of the sonication bath on virus infectivity and determined that the applied sonication did not decrease the virus titer (see Appendix A). The viral titers obtained after selective virus detachment form the surface were compared to those of the reference materials (TiO_2_ anatase, steel 316 L).

### 2.5. Measurement of Surface Free Energy (SFE)

Contact angle measurements were performed using a mobile surface analyzer MSA (Krüss GmbH, Hamburg, Germany) equipped with a double-pressure dosing unit for hydrophobicity determination of the surface. Measurements were performed with 2 μL drops and 1 s of equilibration time under constant conditions. Deionized water and diiodomethane (CH2I2) served as test liquids with different polarities to calculate surface energy according to the OWRK (Owens, Wendt, Rabel, and Kaelble) method [52,53,54]. Surface free energy (SFE) as well as polar and dispersive components were calculated using Krüss Advance software v. 1.9.2.

### 2.6. Statistical Analysis

Differences among disks surfaces were analyzed with ANOVA test. A *p*-value less than 0.05 was considered statistically significant. Statistical analysis was performed using SPSS version 28. Calculations and ANOVA tables are presented in Appendix A).

## 3. Results

In all experiments, we arbitrarily used a 4 logs reduction of virus titer as a measuring stick for efficient inactivation of virus [55,56]. This implied 99.99% inactivation of the viruses.

### 3.1. Virus Inactivation in Metal Ion Salt Solutions

The viability of Phi6 after treatment with various metal salt solutions at different concentrations is shown in Figure 2. Inactivation corresponding to 99.99% was reached with 1 mM silver nitrate and ferrous sulphate, 10 mM aluminum sulphate, and 20 mM copper sulphate (Figure 2A–D). Potassium dichromate and nickel sulphate had lower inactivation efficiencies; at the highest concentration tested (0.1 M), the virus titer decreased by approximately 3 or 1 log, respectively (Figure 2E,F). Since metal salt solutions are known to undergo hydrolysis, we measured the pH of the resulting solutions. In the case of iron and aluminum sulphate, a significant decrease in pH was measured with an increasing salt concentration. In these solutions, the pH decreased below pH 4, which could lead to a drastic inactivation of the virus (Appendix A). In other metal ion solutions, the decrease of pH was less pronounced, or the solution pH did not change (i.e., silver nitrate). Compared to other physicochemical factors such as temperature, pH, relative humidity (RH), sonication, and UV radiation (Appendix A), metal salts of Ag, Cu, Fe, and Al could reach similar virus deactivation rates.

### 3.2. Virus Inactivation in Contact with Ceramic and Metal Powder Suspensions

The virucidal effects of metal and ceramic powder suspensions are shown in Figure 3. Treatment with the TiO_2_–ZrO_2_–Ag–Cu powder suspension had the strongest effect on Phi6 viral titer. Within the experimental error, 99.99% inactivation of the virus was achieved at a metal dopant concentration of 10 mM Ag and Cu ions (Figure 3A). The decrease of virus titer correlated with a decrease in pH. To obtain 4 logs decrease for the TiO_2_–Ag–Cu powder, a 25 mM concentration of Ag in Cu ions was needed (Figure 3B). At higher metal ions concentration, the number of infective virus particles decreased, although the pH remained unchanged. The metallic Co28Cr6Mo powder suspension (Figure 3C) had no significant effect on viral inactivation.

### 3.3. Virus Inactivation on Disks Surfaces

The inactivation of viruses on ceramic and metal composite disk surfaces was determined selectively. The inactivation of viruses suspended in a drop of liquid on disk surfaces is presented in Figure 4A, the inactivation of weakly attached viruses in contact with the disk surfaces in Figure 4B, and the inactivation of strongly attached viruses in Figure 4C. Viruses suspended in a drop of liquid in contact with the control steel disks had comparable viral titers to those of the control. However, there was significant virus inactivation in the drop of liquid in contact with TiO_2_ anatase disks. A more pronounced virus inactivation was observed in droplets collected from TiO_2_ + 10%Ag + 10%Cu disks (1.5 log decrease, *p* < 0.001), whereas a dramatic decrease of liquid volume was observed on 45%TiO_2_ + 5%Ag + 45%ZrO_2_ + 5%Cu ceramic disk and Co28Cr6Mo metal disks. The two disks adsorbed all the added virus suspensions during the 15 min incubation (Appendix A), and no virus suspension was left to test for viral titer (Figure 4A). The infiltration of the added volume of virus suspension was particularly fast (3–5 s) in Co28Cr6Mo metal disks.

Virus contact inhibition by different surfaces was tested by allowing viruses to adhere to the disk surfaces for 15 min after the liquid was removed. The weakly attached viruses were detached by swabbing the surfaces with wet and dry swabs. This simulated touching the surface with moist and dry fingers and possible virus transmission from the surface. The numbers of infective viruses recovered per control steel disk were in the order of 10^7^ PFU/disk. Slightly lower numbers of infective viruses were determined on TiO_2_ anatase ceramic surfaces. On the other hand, viral titers obtained from swabbing TiO_2_ + 10%Ag + 10%Cu, 45%TiO_2_ + 5%Ag + 45%ZrO_2_ + 5%Cu, and Co28Cr6Mo surfaces were significantly lower compared to those from control TiO_2_ or steel disks (*p* < 0.001) (Figure 4B).

Although the majority of the swabbed viruses were detached from the cotton swabs and resuspended into PBS solution, some viruses remained attached to the swabs. This would lead to underestimation of the number of weakly attached viruses to a disk surface. To test this, we put swabs after resuspension in PBS on agar plates (Figure 4D). As expected, a substantial number of residual viruses remained attached to the swabs. The highest numbers of residual viruses were detected on swabs from steel and TiO_2_ composite ceramic disk surfaces. On swabs collected from TiO_2_ + 10%Ag + 10%Cu, 45%TiO_2_ + 5%Ag + 45%ZrO_2_ + 5%Cu, and Co28Cr6Mo disks, considerably less residual infectious viruses were present (seen as individual plaques).

To obtain the number of strongly attached viruses, the disks were put in a sonication bath after swabbing. The recovery of strongly attached viruses that were infective is shown in Figure 4C. From the control steel and TiO_2_ disks, we recovered 5.0 × 10^4^ strongly attached infective viruses. Only a few strongly attached infective viruses were recovered from TiO_2_ + 10%Ag + 10%Cu and 45%TiO_2_ + 5%Ag + 45%ZrO_2_ + 5%Cu disks. The highest viral recovery was obtained from Co28Cr6Mo disks (4.3 × 10^5^).

Surface free energy (SFE) can have a major effect on virus adhesion [57]. The SFE of different surfaces varied between 33.19 (± 1.38) mJ/m^2^ and 55.20 (± 2.76) mJ/m^2^ (Table 1). The highest SFE was measured on the surface of porous Co28Cr6Mo, which also had the highest polar component. On all disks, the disperse component was the dominant component of SFE.

## 4. Discussion

The interactions of phage particles with abiotic surfaces are usually understudied in the field of phage ecology. In this work, we examined the resistance of phage Phi6 to various physicochemical parameters, metal salts, ceramic powders, as well as composite ceramic and metal surfaces fabricated by spark plasma sintering (SPS) and/or selective laser melting (SLM) technology. The results showed that phage infectivity can be quickly decreased (during 15 min) by more than 4 logarithms in contact with various metal ions; this is generally considered a very good virucidal activity [55,56]. More importantly, phage infectivity could decrease significantly on ceramic and metal surfaces doped with metals, which can provide new materials to control viruses.

The results imply that water solutions of Ag, Cu, Fe, Al, and Cr ions have a virucidal effect on Phi6. The loss of Phi6 infectivity associated with iron and aluminum sulphates is likely the consequence of a low pH due to metal ion hydrolysis [58]. For example, at a 10 mM concentration of iron and aluminum sulphates, the pH was below 4, with a significant detrimental effect on phage infectivity. On the other hand, the virucidal effect of Cu and Ag ions is well known [19,20,21,22,29,30,33]. It is interesting to note that the virucidal effect of silver nitrate was obtained without a significant pH drop. On the other hand, the virucidal effect of copper sulphate likely derived from a combination of Cu ions toxicity and low pH values. A concentration of 20 mM copper sulphate was required to achieve a 99.99% loss of Phi6 infectivity. This was higher than reported by Li et al. [21], who showed that 5 mM copper sulphate reduced Phi6 virus titer by 4 logs, albeit after 180 min of incubation. When powder suspensions of TiO_2_ + 10%Ag + 10%Cu were tested for virucidal activity, the activity was lower compared to that of pure ion solutions, suggesting lower ion activities in powder suspension compared to solutions. Still, the powder suspensions were able to reduce viral titers for more than 4 logs. In powder suspensions where TiO_2_ was partially replaced with ZrO_2_, the virucidal activity was lower than in the TiO_2_ + 10%Ag + 10%Cu suspension. On the other hand, in the Co28Cr6Mo powder suspension, there was no significant virucidal activity, suggesting that Co, Cr, and Mo ions have low virucidal activity. Consistently, we obtained appreciable Cr virucidal activity only at high ion concentrations (>100 mM).

In this work, new virucidal composite ceramic and metal materials were fabricated by selective laser melting and spark plasma sintering technology. Most notably, 45%TiO_2_ + 5%Ag + 45%ZrO_2_ + 5%Cu and Co28Cr6Mo disks have porous features that allow the infiltration of the added virus particles. The infiltration on Co28Cr6Mo was completed in less than 5 s. The polar component of surface free energy of Co28Cr6Mo was very high, which may explain the ease with which water infiltrates the disk. Co28Cr6Mo is a pre-alloyed spherical-shaped powder, which is suitable for both SPS and SLM processes [59]. Its excellent biocompatibility and heat and oxidation resistance make it a great candidate for medical industry. The results of this study suggest that this material also has excellent virus infiltration and adsorption properties. The viruses strongly adhered to the interior of the material but could be removed with sonication. It is expected that sonication released viruses from the pore space. When PFU of viruses attached to the internal and external surfaces of Co28Cr6Mo disk were added together, the total number of PFU was still significantly lower than that in control steel disks, suggesting virucidal activity. On the other hand, the infiltration of a drop of viruses in 45%TiO_2_ + 5%Ag + 45%ZrO_2_ + 5%Cu disk must have a different origin. The dispersive contribution to surface free energy was much higher. It is interesting to note that in TiO_2_-based ceramics, there was no infiltration. This suggest that the addition of bioinert ZrO_2_ ceramic increased the porosity of TiO_2_–ZrO_2_ composite ceramics. The mechanical properties of TiO_2_–ZrO_2_ mixtures are influenced by the amount of TiO_2_ and the sintering temperature, due to the lower melting point and density of TiO_2_ compared to ZrO_2_ [60]. However, compared to anatase TiO_2_, the ZrO_2_ nanoparticles have more photocatalytic degradation capability, and the TiO_2_–ZrO_2_ ceramic composite has improved phase stability [61].

The newly fabricated 45%TiO_2_ + 5%Ag + 45%ZrO_2_ + 5%Cu ceramic material showed combined virus adhesion, virucidal, and infiltration capability. The number of infective virus particles in this ceramic was significantly lower compared to that in the reference TiO_2_ ceramic, which does not have inherent virucidal properties, although viruses can adhere to a TiO_2_ surface (Figure 4B,C) [62,63,64]. If one assumes that the same number of viruses in a drop of virus suspension sedimented to the disk surface, irrespective of the surface chemistry, then the decreased number of infective viruses on fabricated surfaces indicates a virucidal contact effect of the new material. The virucidal effect is attributed to Ag and Cu doped ions. When 10^8^ PFU were added to the 45%TiO_2_ + 5%Ag + 45%ZrO_2_ + 5%Cu disk surface, most viruses infiltrated the disk. Approximately 10^4^ virus PFU remained weakly attached to the external surface and could be swabbed, whereas 10^2^ PFU were strongly attached. This suggests that 99.99% of viruses that were added to the surface were either irreversibly attached to the surface or inactivated, therefore possessing no threat to potential host cells.

Phage attachment to different surfaces and their inactivation can have a dramatic effect on virus reproduction success. Bacteriophages modify microbial communities by lysing hosts, transferring genetic material, and effecting lysogenic conversion. All this can be severely compromised when phage numbers decrease due to virucidal effectors in the environment or irreversible attachment to a surface. In addition, when phages are infiltrated in a material, they may fail to find host cells. For example, in a porous soil environment, up to 30% of volume available for viruses can be inaccessible for bacteria in the presence of a pore diameter less than 1 µm [65]. Similarly, attached viruses will be inaccessible for bacteria. Nevertheless, infiltrated or attached phages may retain their viability. In contrast, when phages are in contact with virucidal surfaces such as Cu- and Ag-doped ceramic surfaces, their infectivity can be drastically reduced.

## 5. Conclusions

Understanding phage stability in the environment is important as it determines phage ecology and phage reproductive success. Irreversibly attached phages, inactivated phages, or phages that enter pore spaces smaller than the size of bacterial hosts can no longer infect their bacterial host cells. We showed that inactivation of Phi6 in solutions with Ag and Cu ions can be as effective as inactivation by other environmental factors, such as pH, temperature, or UV. Adding Ag and Cu to powder suspensions decreases their virucidal effect. The new fabricated metal and ceramic surfaces with doped Ag and Cu ions allow very good virus infiltration and have virucidal activity, in particular 45%TiO_2_ + 5%Ag + 45%ZrO_2_ + 5%Cu, which in addition to promote irreversible virus adhesion, also has virucidal activity and favors virus infiltration.

## Figures and Tables

**Figure 1 viruses-14-00204-f001:**
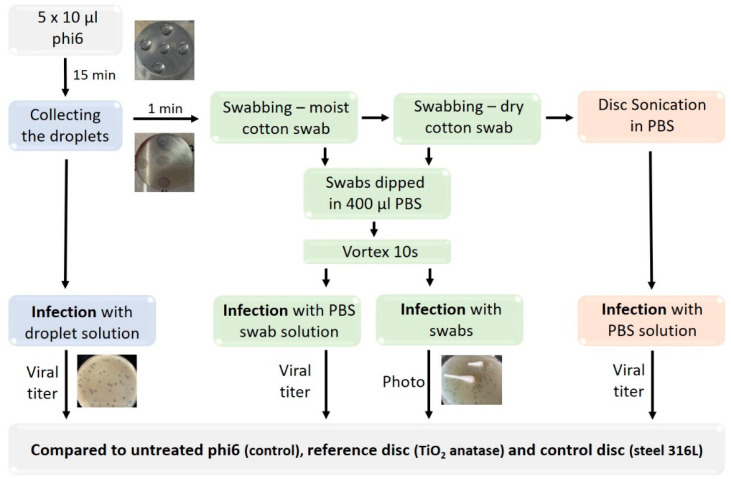
Schematic representation of disk testing.

**Figure 2 viruses-14-00204-f002:**
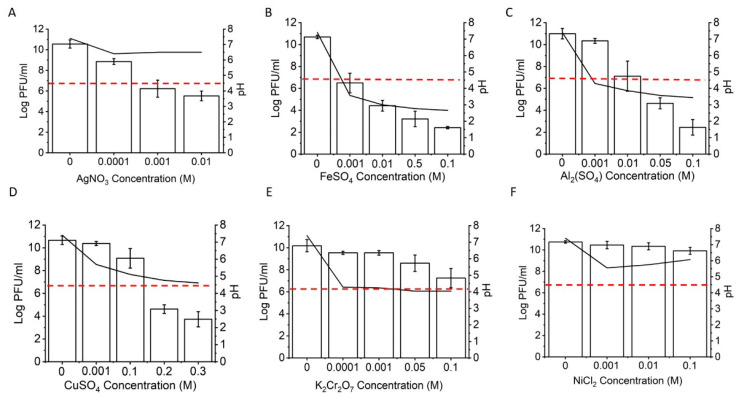
Phi6 viral titer after treatment with metal solutions. (**A**) Phi6 treated with silver nitrate; (**B**) ferrous sulphate; (**C**) aluminum sulphate; (**D**) potassium dichromate; (**E**) copper sulphate; (**F**) nickel chloride. On the left y-axis, PFU/mL, and on the right y-axis, pH of the solutions are presented. The black lines represent the pH of the solutions, the red dashed lines indicate the decrease in viral titer by 99.99% compared to the control. The average values and standard deviations are shown (n ≥ 3).

**Figure 3 viruses-14-00204-f003:**
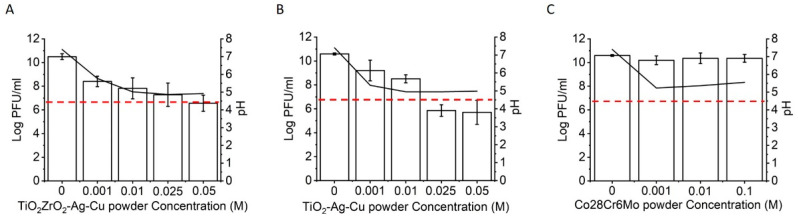
Phi6 viral titers after treatment with metal/ceramic powder suspensions. (**A**) Phi6 treated with TiO_2_–ZrO_2_–Ag–Cu powder; (**B**) TiO_2_–Ag–Cu powder; (**C**) Co28Cr6Mo metal powder. Black lines represent the pH of the ceramic–metal suspensions, and red dashed lines indicate the decrease in viral titer by 99.99% compared to the control. The average values and standard deviations are shown (n ≥ 3).

**Figure 4 viruses-14-00204-f004:**
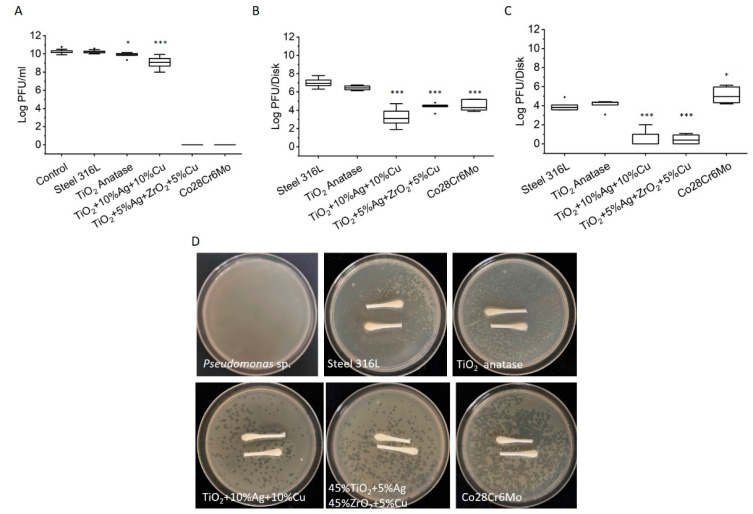
Strongly and weakly attached viruses to ceramic and metal surfaces. Five-digit summary display (minimum, first quartile, median, third quartile, maximum). The circles illustrate the outliers. (**A**) Infective viruses suspended in liquid droplets on disk surfaces; (**B**) weakly attached infective viruses in contact with the disk surfaces; (**C**) strongly attached infective viruses. One asterisk designates *p* < 0.05, and three asterisks *p* < 0.001 significance. (**D**) residual infective viruses on swabs. When no virus particles were present on swabs, *Pseudomonas* sp. could grow and form confluent biofilms on agar plates. When residual viruses were present on swabs, a clear zone of no bacterial growth or individual plaques scattered around the plate were visible.

**Table 1 viruses-14-00204-t001:** Surface free energy (SFE) of different fabricated surfaces.

	SFE (mJ/m^2^)	Disperse (mJ/m^2^)	Polar (mJ/m^2^)
**Steel**	33.19 ± 1.66	30.44 ± 1.52	2.75 ± 0.14
**TiO_2_ anatase**	39.33 ± 1.97	32.26 ± 1.61	7.07 ± 0.35
**TiO_2_ + 10% Ag + 10% Cu**	40.47 ± 2.02	32.81 ± 1.64	7.66 ± 0.38
**45%TiO_2_ + 5% Ag + 45%ZrO_2_ + 5% Cu**	33.77 ± 1.69	30.08 ± 1.50	3.69 ± 0.18
**Co28Cr6Mo**	55.20 ± 2.76	28.58 ± 1.43	26.62 ± 1.33

## Data Availability

Raw data and measurements are available from the corresponding author, upon reasonable request.

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
