# Peer review of "Phi 6 Bacteriophage Inactivation by Metal Salts, Metal Powders, and Metal Surfaces"

_viruses, 2022, doi:10.3390/v14020204_

Round 1

Reviewer 1 Report

The manuscript is well written. Only a few small things should be taken into account.

Line 59: Ref 45 is not suitable. No relation to composites with TiO2 Ag could be found. Therefore, reference should be replaced or removed.

Line 88: Which Leibniz-Institute is meant?

Line 100: Doi is not appropriate.

Line 182, 183: Owens and Wendt, 1969; Kaelble, 1970; are these references? Please include it in the reference section.

Author Response

Reviewer 1:

The manuscript is well written. Only a few small things should be taken into account.

Line 59: Ref 45 is not suitable. No relation to composites with TiO2 Ag could be found. Therefore, reference should be replaced or removed. Rahmani, R., Antonov, M., Kollo, L., Holovenko, Y., Prashanth, K.G. Mechanical Behavior of Ti6Al4V Scaffolds Filled with CaSiO3 for Implant Applications, Applied Sciences. 2019, 9, 3844. https://doi.org/10.3390/app9183844

In this reference we have used the same SLM-SPS combination as in the manuscript. However, Titanium dioxide was replaced by Wollastonite (for in-vivo applications). As this is not directly related to this work we have removed the reference 45, and changed references accordingly.

After authenticate check for your manuscript, we found some sentences are similar with published papers. In order to keep the novelty of your manuscript, we kindly suggest you rephrase the highlighted sentences in your manuscript. “For example, enteric viruses were shown to survive on wool blankets for the longest period of time, poster card for the shortest period of time, and cotton fabric for an intermediate period of time [46,47]. A recent study has found that SARS-CoV-2 can persist longest on propylene plastic surfaces and stainless steel, with infective viruses found up to 72 hr after initial application though at a greatly reduced viral titer [48]”.

 The text has been rephrased and a reference to the original work was added. The new text reads as: “For example, stability of enteric viruses in contact with different soft surfaces was affected by surface type and environment conditions [45,46]. A recent study has found that SARS-Cov-2 was most stabile on polypropylene plastic and stainless steel surfaces (up to 72 hours after application), while on copper no infectious SARS-Cov-2 were detected after 4 hours [27].

Line 88: Which Leibniz-Institute is meant?

This has been corrected in the new version. Leibnitz-Institut, DSMZ-Deutsche Sammlung von Mikroorganismen und Zellkulturen GmbH.

Line 100: Doi is not appropriate.

We have removed the DOI identification number.

Line 182, 183: Owens and Wendt, 1969; Kaelble, 1970; are these references? Please include it in the reference section.

These references are now being included in the reference list.

New references added:

  1. Kaelble, D. H. Dispersion-Polar Surface Tension Properties of Organic Solids. J. Adhesion 1970, 2, 66‒81
  2. Owens, D., Wendt, R. Estimation of the Surface Free Energy of Polymers. In: J. Appl. Polym. Sci 1969, 13, 1741‒1747
  3. Rabel, W. Einige Aspekte der Benetzungstheorie und ihre Anwendung auf die Untersuchung und Veränderung der Ober-flächene genschaften von Polymeren. Farbe und Lack 1971, 77, 997‒1005

Reviewer 2 Report

The work of K. Molan et al. entitled “Phi 6 bacteriophage inactivation by metal salts, metal powders, and metal surfaces” presents interesting and useful subject of study. The manuscript was prepared clearly and logically. A few mistakes and shortcomings, which I found, do not diminish the value of presented research. All my comments are listed below.

Particular objections.

  1. Introduction, line 75: please, correct the notation of chemical formula.
  2. Chapter 3.1, line 195: please, check the number of Figure.

In summary, in my opinion the paper should be corrected, taking into consideration the above suggestions. Then the paper may be recommended for publication.

Author Response

Reviewer 2:

The work of K. Molan et al. entitled “Phi 6 bacteriophage inactivation by metal salts, metal powders, and metal surfaces” presents interesting and useful subject of study. The manuscript was prepared clearly and logically. A few mistakes and shortcomings, which I found, do not diminish the value of presented research. All my comments are listed below.

Particular objections.

Introduction, line 75: please, correct the notation of chemical formula.

The formula has been corrected.

From Al2(SO4)3 to Al2(SO4)3

Chapter 3.1, line 195: please, check the number of Figure.

Thank you for spotting the lapse in citing the Figure. This has been now corrected to: ”The viability of Phi6 after treatment with various metal salt solutions and concentrations is shown in Figure 2.”